# Inspiratory Muscle Training in Patients with Chronic Obstructive Pulmonary Disease (COPD) as Part of a Respiratory Rehabilitation Program Implementation of Mechanical Devices: A Systematic Review

**DOI:** 10.3390/ijerph19095564

**Published:** 2022-05-03

**Authors:** Eva Vázquez-Gandullo, Antonio Hidalgo-Molina, Francisca Montoro-Ballesteros, María Morales-González, Isabel Muñoz-Ramírez, Aurelio Arnedillo-Muñoz

**Affiliations:** 1Pneumology, Allergology and Thoracic Surgery Department, University Hospital Puerta del Mar, 11009 Cádiz, Spain; antoniohmolina@hotmail.com (A.H.-M.); paquimb_88@hotmail.com (F.M.-B.); isabel.murami@gmail.com (I.M.-R.); 2Pneumology Department, Hospital Punta Europa, 11207 Algeciras, Cádiz, Spain; moralesgonzalez.maria@gmail.com

**Keywords:** inspiratory muscle training, inspiratory restriction device, chronic obstructive pulmonary disease, respiratory rehabilitation, quality of life

## Abstract

Chronic Obstructive Pulmonary Disease (COPD) is a complex and heterogeneous disease, with pulmonary and extrapulmonary manifestations, which leads to the need to personalize the assessment and treatment of these patients. The latest updates of national and international guidelines for the management of COPD reveal the importance of respiratory rehabilitation (RR) and its role in improving symptoms, quality of life, and psychosocial sphere of patients. Within RR, the inspiratory muscle training (IMT) has received special interest, showing benefits in maximum inspiratory pressure, perception of well-being, and health status in patients with chronic heart disease, respiratory diseases, and dyspnea during exercise. The aim of this review is to assess the efficacy of IMT in COPD patients through the use of inspiratory muscle training devices, compared with respiratory rehabilitation programs without inspiratory muscle training. In the last years, many mechanical devices focused on inspiratory muscle training have been developed, some of them, such as the AirOFit PRO™, PowerBreath^®^, or FeelBreathe^®^, have shown clear benefits. The active search for candidate patients to undergo the RR program with inspiratory muscle training using this type of device in COPD patients represents an advance in the treatment of this disease, with direct benefits on the quality of life of the patients. In this article, we review the available evidence on IMT in these patients and describe the different devices used for it.

## 1. Introduction and Objectives

Chronic Obstructive Pulmonary Disease (COPD) is a preventable and treatable disease, characterized by the presence of persistent respiratory symptoms and airflow obstruction [1]. Currently, this disease is one of the first causes of worldwide morbimortality [2]. The best known risk factor is exposure to tobacco smoke, although other factors such as environmental pollution or exposure to biomass combustion, together with individual predisposition, have also been correlated [1]. It is a complex and heterogeneous disease, with pulmonary and extrapulmonary manifestations, which leads to the need for a comprehensive approach to it [3,4]. The variability between individuals and within the same individual is ostensible, the cardinal symptom being dyspnea, frequently accompanied by cough and expectoration [5]. These symptoms are related not only to bronchial obstruction, but also to deconditioning, which leads to a decrease in physical activity. The air trapping and dynamic hyperinflation present in these patients are associated with an overload at the muscular level, which ends up conditioning a vicious circle that is difficult to break [6].

The guidelines for the management of COPD highlight the importance of respiratory rehabilitation (RR) and its role in improving symptoms, quality of life, and in the psychosocial sphere. Understanding this intervention is akin to a global evaluation of health status, followed by therapies adjusted to individual needs, including exercise training, education, and behavioral therapy [1,7,8]. RR is postulated as being the most cost-effective treatment strategy [8]. The implementation of respiratory rehabilitation in patients with COPD is widely accepted, demonstrating a reduction in hospital admissions and mortality in patients with frequent exacerbations [9]. Inspiratory muscle training (IMT) has received special interest as part of RR. In the 1970s and 1980s, Rochester [10] and Chen [11] highlighted inspiratory muscle weakness in COPD patients and the potential benefits of targeted training. Posterior studies have confirmed the efficacy of implementing IMT as part of a RR program in a certain profile of patients with COPD, showing improvements in maximum inspiratory pressure, perception of well-being, and health status in patients with chronic heart disease, other respiratory diseases, and dyspnea during exercise [12,13,14].

The aim of this review is to explain inspiratory muscle training, what devices can be used for its implementation, and the studies that support its use in COPD patients. We evaluated the efficacy of IMT in COPD patients with and without inspiratory muscle weakness in a RR program, and finally the usefulness of implementation of inspiratory restriction devices in domiciliary programs.

## 2. Material and Methods

A systematic review has been carried out following PRISMA recommendations [15] for descriptive and systematic reviews. To carry out a structured search, in addition to following PRISMA recommendations, we raised the clinical question to be answered using the PICO (patient, intervention, comparison, outcome) methodology. According to this, we considered: patients with COPD; intervention was assessed as IMT to improve the quality of life, as well as peak inspiratory pressure and breathlessness on exertion; we compared inspiratory muscle training with mechanical devices included in a respiratory rehabilitation program versus a standard respiratory rehabilitation program; the results were measured as improvement in quality of life, as well as other variables like decreased dyspnea during exertion, meters walked in the 6MWT and improvement in inspiratory capacity. The authors assessed PubMed and EMBASE from 1 to 28 February, independently, carrying out the literature search in a stepwise manner, based on title and abstract The search engines were selected by the authors for accessibility and reproducibility. Finally, it was decided to continue with satisfactory results, although without overlooking the limitation that this search strategy entails. The following keywords were included in the search, alone or in combination: “inspiratory muscle training”, “inspiratory restriction device”, “chronic obstructive pulmonary disease”, “respiratory rehabilitation”, and “quality of life”. The main search yielded a result of 413 posts. Studies were included according to the following criteria: (a) systematic review, meta-analysis, and original articles; (b) published in the last 10 years; (c) efficacy of the inspiratory muscle devices; (d) studies evaluating the effects of inspiratory muscle training with a threshold device in stable or experiencing acute exacerbation COPD patients. Articles meeting exclusion criteria, e.g., do not specifically deal with IMT, low methodological quality, duplicate studies, lung diseases other than COPD, and studies carried out in active athletes in competition, were eliminated. Finally, 16 documents were considered eligible to evaluate IMT devices (Figure 1). To find other relevant articles, we also assessed the reference list of the collected papers. Two of the biases of the search strategy would be (1) it was carried out independently among the authors, and (2) the use of only two search engines. Nonetheless, the results allow us to answer the established PICO question.

Table 1 shows the analyzed studies for the IMT devices.

## 3. Results

Sixteen articles met the search criteria indicated above and were included. Of the initial 416 found by the search engine, 286 were excluded because they did not perform an IMT intervention, 35 studies did not have the required methodological quality, 23 were duplicate records and other reasons to exclude reports were: other non-COPD lung diseases, and studies in competitive athletes.

### 3.1. Profile of COPD Patients Who Are Candidates for IMT What Is a RR Program with IMT?

COPD patients present a limitation in their activity due to peripheral muscle dysfunction, which, joined to functional alterations in ventilation and gas exchange, produce an increase in dyspnea and fatigue on exertion. As a consequence, they tend to have a sedentary lifestyle and decreased mobility, which further contributes to muscle dysfunction and physical deconditioning, increasing morbidity and mortality in these patients, compared to healthy adults [16].

Dyspnea is the main symptom in COPD patients, which origin may be multifactorial, involving physiological, psychosocial, and environmental spheres, which complicates its approach because the treatment of these symptoms may vary depending on the mechanism that originates it [16]. In addition, the intensity of the symptom does not correlate with the severity of the bronchial obstruction, although it is related to disease progression in cases of severe dyspnea and is usually associated with a decrease in the quality of life and survival at 5 years [17].

Within the pathophysiology related to dyspnea in COPD patients, it has been observed that during intense exercise, the respiratory muscles can consume up to 16% of oxygen, mainly for the skeletal muscles [18,19]. Therefore, if the respiratory muscles’ need for oxygen is very high, a competitive demand will be generated with respect to the active skeletal muscles, which could limit their oxygen supply. In the long run, this situation leads to an increase in the respiratory rate, with a decrease in expiratory time and air trapping, which implies a decrease in inspiratory capacity carrying to a greater sensation of dyspnea, regardless of the expiratory limitation at flow. This sustained situation favours respiratory muscle fatigue, which could alter the optimal ventilation required [6,20], and limit physical performance. McConnelly and Sharpe [21] in their research on healthy subjects concluded that respiratory muscles are resistant to fatigue in non-pathological conditions and at rest, but during intense and prolonged exercise it is possible to reach fatigue of the respiratory muscles. This study reveals that the main respiratory factors limiting physical exercise are the energetic compromise of the respiratory muscles with respect to the active skeletal muscles and the fatigue, so it is reasonable to believe that there may be improvements derived from training respiratory muscles.

Taking into account all of the above, the implementation of an IMT program as part of RR may be indicated in patients with COPD and inspiratory muscle weakness with the aim of improving symptoms, especially dyspnea, and quality of life, although the mechanisms by which this improvement is achieved are unclear [14,22]. Among the authors who have confirmed the beneficial effects of an IMT program joined to RR, Gosselink et al. [13] observed benefits in exercise capacity, including patients with low PaO2 or high PaCO2, improving in both cases after the training of the inspiratory muscles. In the IMTCO study, published by Charususuin, they observed an improvement in the 6 min walking test (6MWT), the primary objective, with the use of the POWERbreathe KH1 device for the IMT. In addition to this, they found an improvement in inspiratory muscle function, health-related quality of life, and daily physical activity in patients undergoing this type of training [22]. Weiner et al. published that inspiratory threshold loading training, added to general exercise reconditioning, markedly improved inspiratory muscle strength and endurance, as well as exercise tolerance, in patients with COPD, and that the improvement in this group of patients was significantly greater than that achieved with general exercise reconditioning alone [23]. In this direction, Petrovic et al. [12], showed the benefits of IMT in patients with dynamic hyperinsuflation, as it produces an increased end-expiratory lung volume that may result in a greater weakness of the inspiratory muscles. They found an increase in exercise capacity and significant reduction in dynamic hyperinflation. In the same way, in 2018, a study was published that demonstrated a positive association between a 8 weeks IMT program with an increased capacity to sustain high ventilation for a longer duration, accompanied by consistent improvements in diaphragmatic strength [24]. This program also consisted of strength exercises followed by endurance ones.

### 3.2. What Is a Respiratory Rehabilitation Program with Inspiratory Muscle Training?

Prior to start a rehabilitation program, it is advisable to demonstrate the existence of inspiratory muscle weakness (PImax), and thus add specific exercises for IMT, aimed at improving physical exercise performance, through repeated resistance exercises [25], which causes an improvement in lactate clearance kinetics and a decrease in the perception of the effort made [26].

To carry out this type of training, devices have been designed and are commonly used in rehabilitation programs, improving respiratory muscle strength and endurance, resulting in a decrease in dyspneic sensation and an increase in exercise tolerance [27]. Three types of devices are available for inspiratory muscle training: threshold devices, resistive load devices, and voluntary isocapnic hyperpnea devices [28].

Threshold device: training with this type of device is obtained with a hand-held device that allows airflow during inspiration after reaching an inspiratory pressure. The effort required by the inspiratory muscles can be adjusted by the tension of a spring; this tension determines the opening of the valve;Resistive loading device: this is one of the most commonly used categories. In this category we have different devices. The PFLEX resistive Trainer device (Respironics HealthScan Inc., Cedar Grove, NJ, USA), consists of a mouthpiece and a circular dial. Turning the dial varies the size of the opening through which the patient breathes. The smaller the opening, the greater the resistance to inspiration. It has 6 diameter sizes. The objective of this exercise is to increase the load on the inspiratory muscles progressively. Many studies have used this device for IMT [25,29]. The PowerBreathe^®^ device also stands out for its widespread use [30]. The Feelbreathe^®^ device also behaves like a resistive load device, but in this case it is nasal and not buccal. It is a nasal ventilatory flow restriction device composed by a strip of hypoallergenic material that is placed and adhered under the nostrils, provoking resistance to flow. Depending on the size and porosity of the device material, the inspiratory process is more or less difficult. This device has the possibility of using it not only in a static situation but also dynamically during exercise [31];Voluntary isocapnic hyperpnea device: consisting of a device that increases the ventilation level of the subject to a predetermined level. The increase in ventilation causes an increase in respiratory rate, which can reach 50–60 rpm. This type of respiratory muscle training requires the patient to perform prolonged periods of hyperpnea, lasting up to 15 min and with a frequency of twice a day, 3 times a week, for 4–5 weeks [32]. To avoid hypocapnia, exercise should be performed on an isocapnic circuit, which maintains stable CO_2_ levels. One device using this method is the SpiroTiger^®^ (Ideag Lab, Ziirich, Switzerland).

Respiratory rehabilitation and IMT programs that complete the recommended time of 6 to 12 weeks have shown, as mentioned above, benefits in health-related quality of life, improvement of dyspnea and exercise tolerance [32]. The main objective is to achieve a change in the patient’s lifestyle habits, avoiding sedentarism, and incorporation to physical activity. Regarding the duration of the training program, it should be taken into account that a short intervention implies an increase in inspiratory muscle strength while a longer intervention increases functional capacity. The benefits last approximately 12–18 months if habitual activity is abandoned, returning the patient to his or her previous situation before starting respiratory rehabilitation.

## 4. Inspiratory Muscle Training Programs Based on the Use of Mechanical Devices

Mechanical devices have shown to improve muscle strength and endurance in patients with COPD, leading to an improvement in the quality of life of these patients [13,33]. However, it has not been demonstrated that there is any additional benefit to an isolated pulmonary rehabilitation program without inspiratory muscle training [8,33,34]. In fact, its implementation in RR programs in COPD patients in general is not recommended by the British Thoracic Society (BTS) or by the Spanish Society of Pneumology and Thoracic Surgery (SEPAR), although the latter recommends adding IMT for patients with inspiratory muscle weakness, defined as a maximal inspiratory pressure of less than 60 cmH_2_O [35,36]. This variety of training usually employs small devices that are easily manipulated by the patient. Interest in this subject has been progressing over the last half century, reflecting the increase in the number of publications, which may be related to technological advances. These studies use simple devices, which generally increase inspiratory resistance by gradually decreasing the inspiratory orifice, showing a benefit for inspiratory muscle training in patients with COPD. As a main limitation of these studies, the sample size included is small [37,38,39]. Dekhuijzen et al. [40] compared a respiratory rehabilitation program to which they associated IMT with a flow resistance device, versus RR, observing an improvement in physical exercise capacity measured by the distance covered in a 6MWT and the maximal oxygen consumption (VO_2_max), in those patients in whom they followed a muscular inspiratory training program associated with the RR program.

Some IMT devices are currently available and are described below (Figure 2).

### 4.1. Respifit S^TM^

Respifit S^TM^ is a threshold type device for IMT that is portable, small, and easy to use. It features a “Y” shape, mouthpiece, nasal closure clip, and a display that shows results and facilitates therapy monitoring. The mode of use consists of initially performing normal inhalations and exhalations, after which the subject should inhale and exhale slowly and deeply, without hyperventilating. The subject then makes slow turns of the head to one side and inhales and exhales, and then to the other side, repeating breaths, for one minute. The next minute, he performs the same procedure but with up and down movements of the head. Finally, the subject will lean forward to touch their toes while breathing in and out. Finally, the subject will return to normal inhalation and exhalation.

This device has been studied by Petrovic et al. [12], aiming to analyze the effects of IMT on exercise capacity, dyspnea, and inspiratory fraction during exercise in patients with COPD. A total of 20 patients were included and divided into two groups. Patients in the treatment group performed an inspiratory muscle training program using the Respifit S^TM^ device daily for 8 weeks, with the other group serving as a control. Assessment of exercise capacity was measured by cardiopulmonary exercise test prior to the start of training and one week after the end of training. Improvements in respiratory muscle function, exercise capacity and quality of life were observed, as well as a decrease in dyspnea.

### 4.2. PowerBreathe^®^

PowerBreathe^®^ is an electronic resistive loading device that is small and light with a mouthpiece at the top and a display on which different parameters such as maximal inspiratory pressure (cmH_2_O), maximum inspiratory flow (l/s), training load (cmH_2_O), power (watts), average inhaled volume (l), and T-index (training intensity index) can be observed. Inside the device there is a quick response valve with electronic control to generate inhalation resistance.

The patient should inhale and exhale through the mouthpiece 30 times. This regimen is recommended to be repeated twice a day. When inhaling, the patient will notice a resistance that varies in relation to the volume of air. The training resistance is maximal at the beginning of the inhalation (RV—residual volume) and gradually decreases to values close to zero at the end of the inhalation (TLC—total lung capacity). This resistance is designed to match the length-tension relationship of the inspiratory muscles, providing a constant relative training intensity at all lung volumes.

This training method ensures optimal stimulation throughout the entire range of motion of the inspiratory muscles. The training load is introduced gradually over the first five breaths of a training session. The first two breaths are performed without load. During these breaths, inhaled volume and flow are measured and used to establish an appropriate training load. The training load is then gradually introduced during breaths three and four times until the full load is reached in the following breaths.

The training load is adjustable and should be set at a level appropriate for the patient to effectively train the inspiratory muscles. Research has shown that inspiratory muscle training loads should exceed 30% of the maximal inspiratory muscle pressure (force) to be effective.

Magadle et al. [23], analyzed the effect of adding IMT in patients with severe COPD (FEV_1_ < 50%) without respiratory failure, who were already included in a rehabilitation program, assessing lung function, inspiratory muscle strength, perception of dyspnea, exercise performance, and quality of life. All patients participated in a rehabilitation program for 12 weeks, after which they were randomized into two groups: one group was given IMT with PowerBreathe^®^ and the other was given a sham inspiratory training device (control group). Statistically significant differences were observed in inspiratory strength and in the perception of dyspnea, however, there were no differences between the two groups with respect to FEV_1_ or 6MWT.

The following study also aims to evaluate the added effect of IMT together with full body resistance training. For this purpose, it divides the patients into an experimental group (IMT using the PowerBreathe^®^ device together with full body training) and a control group (full body training alone). After the training period an improvement in the Berg Balance Scale (BBS) was demonstrated and statistically significant differences were also found in inspiratory muscle strength, increasing up to 37% in the experimental group. However, no differences were found between both groups in the 6MWT [41].

Following the same line, Charususin et al. [42] conducted a study with the aim of evaluating the added benefits of IMT to a rehabilitation program in COPD patients with inspiratory muscle weakness. Although a statistically significant improvement in inspiratory muscle strength and endurance was established, this was not reflected in the 6MWT, where no differences were found.

An advantage of the PowerBreathe^®^ device is that it allows unsupervised training. Langer et al. [43] compared the efficacy of a mechanical loading threshold device versus an electronic device for home use, the PowerBreathe^®^. They note that the use of the electronic device with a home training program requires less time investment for the health services, and there is an improvement in inspiratory muscle capacity. In addition, study participants in the electronic device group tolerated higher training intensities and achieved significantly greater improvements in inspiratory muscle function [44].

### 4.3. Threshold IMT^®^

Threshold IMT^®^ is a small and light threshold type device that is cylindrical in shape, with a nozzle at one end. This device incorporates a unidirectional valve independent of the flow, which ensures constant resistance and allows pressure adjustment (in cmH_2_O). This valve provides a resistance to the air flow, which forces the subject to make a greater effort to overcome the pressure [45]. The subject must repeat inspiration and expiration manoeuvres through the mouthpiece, perceiving a variable resistance only in inspiration. A use of about 10–15 min twice a day is recommended.

According to the existing literature, the benefits derived from IMT with mechanical devices are greater in COPD patients with a maximal inspiratory pressure of less than 60 cmH_2_O (13), but there are studies that aim to demonstrate the efficacy of the use of these devices in patients with a maximal inspiratory pressure above 60 cmH_2_O. Beaumont et al. [46], with the IMT Threshold IMT^®^ device, found no statistically significant differences in terms of improvement in dyspnea or functional parameters. No benefits were observed in a study in which two groups were compared: training with cycloergometer plus IMT with Threshold IMTR; versus training with cycloergometer alone. It was observed that combined training improves inspiratory strength versus isolated cycloergometer training; however, no differences were observed with regard to improved physical performance or dyspnea. Another interesting aspect of this study is that an analysis was performed in a subgroup of patients with inspiratory muscle weakness, defined as a maximal inspiratory pressure of less than 60 cmH_2_O, without noticing the benefits of combined training in this subgroup [47].

### 4.4. PrO2Fit TM^®^

PrO2Fit TM^®^ is a small, lightweight, and portable resistive load device with a nozzle on the end. It is connected to a mobile application, which provides the user with a graphic representation of the effort made. For its use, the patient must repeatedly inhale and exhale through the mouthpiece, following the indications of the mobile application.

This device allows us to evaluate the inspiratory musculature by means of an incremental respiratory endurance test, known as TIRE (Test of Incremental Respiratory Endurance). In this way, the maximum inspiratory pressure is measured over time, obtaining the maximum sustained inspiratory pressure.

This device was developed for inspiratory muscle training in athletes and is of interest in patients with respiratory pathology; however, there are few studies that evaluate its efficacy in patients with COPD. One such study, by Formiga et al. [48], evaluates the reliability of such a device for measuring inspiratory muscle strength and endurance, but is not designed to test its efficacy as an inspiratory muscle training device. Although not focused on COPD patients, McCreery et al. [49] evaluated the effect of inspiratory muscle training using the PrO2 Fit device in patients with bronchiectasis, concluding that there is an increase in inspiratory muscle strength and endurance in these patients.

The protocol of a randomized clinical trial comparing a home training program based on an incremental breathing endurance test (TIRE), using the PrO2 device, versus a traditional inspiratory muscle training program using the Threshold IMTR has recently been carried out. The results of this study are not yet published [50].

### 4.5. Aerosure Medic^®^

Resistive charging device designed to provide resistance on inspiration. It consists of a rectangular device with a mouthpiece and a charger. It has two modes, one for muscle training in which the patient must repeatedly inhale and exhale through the mouthpiece and a second mode with a mucolytic effect on the airway.

In the study carried out by Daynes et al. [51], such a device is evaluated. Patients were included in a training program with Aerosure Medic^®^, to be used 3 times a day for a total of 8 weeks. A statistically significant improvement in dyspnea and in maximal inspiratory pressure (PImax) and maximal expiratory pressure (PImax) was observed, with the improvement in PImax being greater in the subgroup of patients with inspiratory muscle weakness, also defined in this study as a PImax of less than 60 cmH_2_O.

### 4.6. AeroFit IMT^®^

AeroFit IMT^®^ is a resistive loading device for IMT, currently under study. It consists of a small, portable, lightweight oral pressure manometer with a rubber rimmed mouthpiece. It contains resistance wheels that provide adjustable airflow. This resistance causes fatigue in the respiratory muscles, which is then compensated by increased muscle mass, making these muscles stronger, faster, and more efficient.

The device is connected via an app to the cell phone where the subject can choose the desired training program and observe the progress achieved through training.

It has been studied in athletes, demonstrating an increase in maximal respiratory strength without reporting dyspnea or respiratory fatigue after use. These results suggest its future application in patients with respiratory diseases such as COPD within an IMT. A clinical trial is currently under development in which the ability of AeroFit IMTR to improve inspiratory muscle strength in COPD patients is being evaluated. The results are not yet published [52].

### 4.7. MicroRPM

MicroRPM is a resistive loading device that is small, portable, lightweight, and noninvasive, containing a mouth-pressure manometer with a rubber flanged mouthpiece. It displays the test results in a device monitor, uses software and calculates the maximal inspiratory pressure and maximal expiratory pressure values (in cmH_2_O), from the one-second average maximum pressure. The MicroRPM needs a different adapter to be adjusted for inhalation and exhalation, without the need for special preparation in terms of cleaning and disinfecting the device by simply adapting the respective inhaler and exhaler adapter and the removable mouthpiece. Stavrou et al. did not find differences between two devices (MicroPRM vs. AirOFit PRO™); the comparison showed no differences in the maximal inspiratory pressure and maximal expiratory pressure variables, but statistically significant differences were observed between the devices in the parameter ease of use and information during the trials [52].

### 4.8. SpiroTiger^®^

This is a voluntary isocapnic hyperpnea device. It consists of a tube that connects to a breathing bag with a mouthpiece at a 90° angle. Between these components there is a side port with an opening that allows inspiration and expiration into the ambient air. It also has a monitor that shows and assists the patient by providing auditory and visual feedback. When the patient exhales, the bag fills with air, with higher concentrations of carbon dioxide. When the bag is full, the side valve opens and allows the remaining exhaled air to escape. Once expiration is complete, the valve closes and gives way to inspiration. During inspiration, the bag is first emptied completely and then the side valve is opened.

A review conducted in 2015 evaluated five articles analyzing the SpiroTiger^®^ device, one of them in patients with COPD and the rest in patients with cystic fibrosis. Finally, they conclude that this device presents an improvement in physical condition, measured in the different studies using the 6MWT and in VO_2_max, as well as an improvement in the quality of life. However, the improvement in FEV_1_ is not conclusive, as it is present in only two of the studies evaluated in the review, and therefore more studies are required [53].

There is a study evaluating the efficacy of SpiroTiger^®^ training for four weeks in patients with COPD. Maximum inspiratory pressure, 6MWT distance, and quality of life were measured using the St George Respiratory Questionnaire for subjects with COPD, and an increase in maximum inspiratory pressure, 6MWT distance, and quality of life were observed [54].

### 4.9. FeelBreathe^®^

So far, the aforementioned devices must be used in a static position and breathing through the mouth, which is not considered physiological. Recently a device for inspiratory muscle training called FeelBreathe^®^ has been designed. It is a nasal ventilatory flow restriction device consisting of a strip of hypoallergenic material that is placed and adhered under the nostrils, producing resistance to flow. Depending on the size and/or porosity of the device material, the inspiratory process is more or less difficult.

The advantage of this device with respect to those mentioned above is, on the one hand, the fact that it is a nasal device, so breathing is more physiological and, on the other hand, it allows for use in dynamic situations, so we can use it while the patient is exercising.

This device has been shown to be effective in improving ventilatory and cardiac efficiency in healthy patients, and in patients with COPD it has also demonstrated less dynamic hyperinflation with better ventilatory efficiency [50]. This ability of the device to be used in motion allows it to be implemented in pulmonary rehabilitation programs by using it during different exercises, and hence the hypothesis that the FeelBreathe^®^ device brings added benefits to respiratory rehabilitation. Gonzalez-Montesinos et al. studied this circumstance and observed that patients who have used FeelBreathe^®^ together with a pulmonary rehabilitation program show an improvement in quality of life, dyspnea, exercise capacity, and inspiratory muscle strength. However, the main limitation of these studies is the small number of patients, so more studies are needed to reach a definitive conclusion on the addition of FeelBreathe^®^ in a pulmonary rehabilitation program [31,55,56].

## 5. Conclusions

The bibliography supports the use of IMT devices, despite the presented limitations: the sample size is low in most of the studies; the compared groups are not clearly standardized—therefore in some of them they may not be comparable, and the follow-up is different in all the studies. Despite the heterogeneity of the studies included in this review limiting the extrapolation of results, a wide benefit in terms of quality of life due to the use of inspiratory muscle training devices has been stated. One limitation of the review process was that the authors undertook an independent review, sharing data afterwards. Another limitation could be the use of just two search engines.

All things considered; it can be concluded that COPD is a heterogeneous disease that requires a multidisciplinary approach for its control. Respiratory rehabilitation is postulated as an important part of non-pharmacological therapeutic options, with IMT taking special interest in recent decades as a part of the rehabilitation program. The IMT has been favoured by the development of different mechanical devices that help both in programs with direct supervision, as well as in others controlled by remote monitoring. Some of the devices that stand out in the chronic obstructive profile patients and present criteria of easier use, accessibility, and good results have been the AirOFit PRO ™, PowerBreathe^®^, or FeelBreathe^®^. This last device has the added value that it is a nasal device, favouring a more physiological breathing, and it also allows its use in dynamic situations, therefore it could be used when the patient is exercising. The active search for candidate subjects to carry out a PR program with training of the inspiratory muscles using this type of device in patients with COPD represents an advance in the treatment of this disease, with direct benefits on the quality of life of these patients. Nonetheless, further studies are needed to reach consensus regarding its standardization in clinical use.

## Figures and Tables

**Figure 1 ijerph-19-05564-f001:**
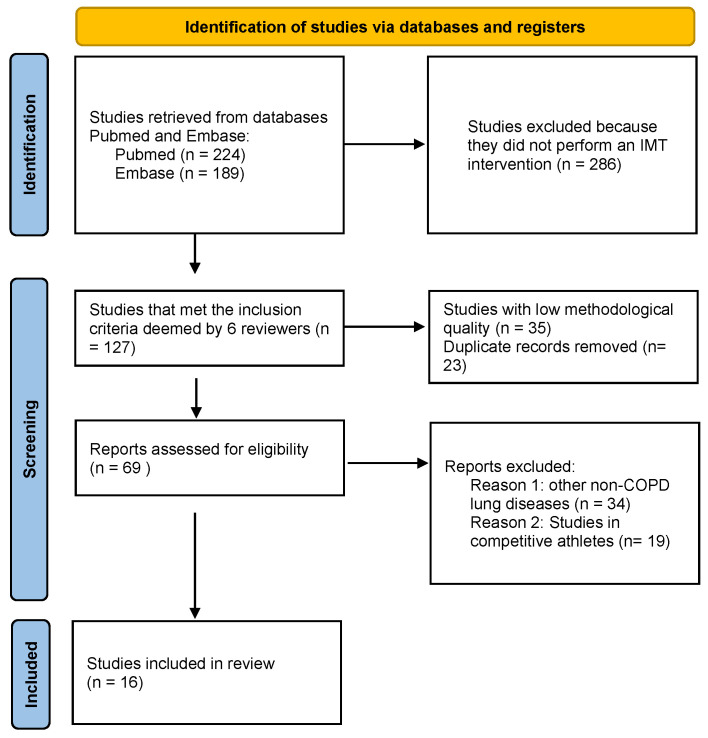
PRISMA 2020 flow diagram.

**Figure 2 ijerph-19-05564-f002:**
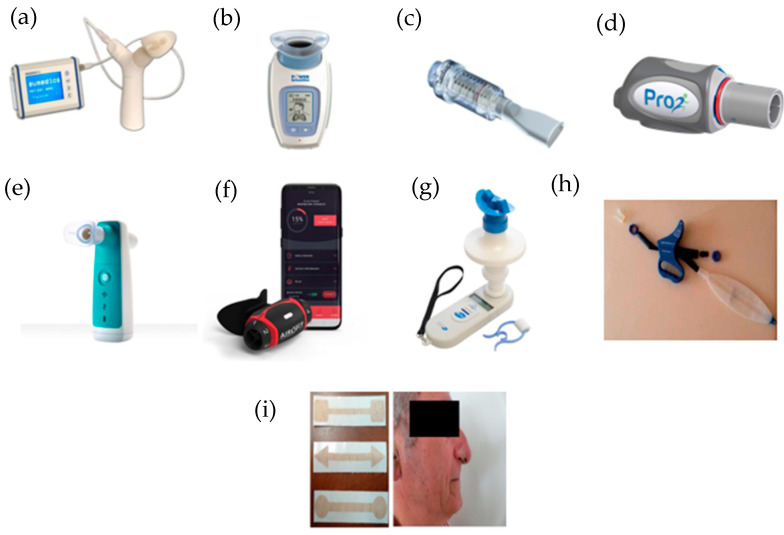
IMT devices used in respiratory rehabilitation. (**a**) Respifit S^TM^; (**b**) PowerBreathe^®^; (**c**) Threshold IMT^®^; (**d**) PrO2Fit TM^®^; (**e**) Aerosure Medic^®^; (**f**) AeroFit IMT^®^; (**g**) MicroRPM; (**h**) SpiroTiger^®^; (**i**) FeelBreathe^®^.

**Table 1 ijerph-19-05564-t001:** Bibliography search of IMT devices.

Author	Publication Date	Device	Type of Devices	Subject Study	Duration	Method	Results
Petrovic et al.	2012	Respifit $S^{\mathrm{TM}}$	Threshold	20 COPD	8 weeks	Direct by cardiopulmonary and stress test	Enhances inspiratory muscle function, dyspnea, and quality of life.
Magadle et al.	2007	PowerBreathe^®^	Resistive load and	34 COPD	12 weeks	6MWT, SGRQ	Enhances inspiratory muscle function and dyspnea perception.
Tounsi et al.	2021	PowerBreathe^®^	Resistive load and	32 COPD	8 weeks	BBS, ABC, 6MWT	Enhances inspiratory muscle function and functional balance according to BBS and ABC.
Charususin et al.	2018	PowerBreathe^®^	Resistive load and	219 COPD	5–8 weeks	6MWT.	No differences in 6MWT. Gains in respiratory muscle function and also endurance exercise capacity.
Langer et al.	2018	PowerBreathe^®^	Resistive load and	20 COPD	8 weeks	PImax, T.lim	Improvments in Pi,max and T,lim. Telemonitorization.
Beaumont et al.	2015	Threshold IMT^®^	Threshold	23 COPD	3 weeks	Borg scale, 6MWT, PImax. Cycle ergometer	Subgroup of patients with FEV_1_ < 50% pred., dyspnea was significantly improved.
McCreery et al.	2021	PrO2 Fit	Resistive load	10 BQ	8 weeks	VO_2_, VCO_2_ and pulmonary function (FVC, FEV_1_)	Increased inspiratory muscle strength and endurance. Telemonitorization.
Formiga et al.	2018	PrO2 Fit	Resistive load	81 COPD	1 day	FEV_1_, FVC, 6MWT, MIP, ID, SMIP.	Increased inspiratory muscle strength and endurance
Formiga et al.	2020	PrO2 Fit	Resistive load	On going	8 weeks	mMRC, FEV_1_, FVC, 6MWT.	Test of incremental respiratory endurance training method has the potential to provide additional clinical benefits in COPD.
Daynes et al.	2018	Aerosure Medic^®^	Resistive load	23 COPD	8 weeks	mMRC, PImax, PEmax. ISWT, ESWT	Improvmente PI max, PE max, and reducing dyspnoea.
Stavrou et al.	2021	AeroFit IMT^®^ MicroRPM	Resistive load	21 athletes	1 day	PSQI, pulmonary function test, ergospirometry	Compare both not differences between devices. AirOFitPRO™ is easier to operate as a device and provides more information.
Bernardi et al.	2015	SpiroTiger^®^	Resistive load	20 COPD	4 weeks	Spirometry 6MWT, VO_2_max, SGRQ	Increased inspiratory muscle and quality of life.
Włodarczyk et al.	2015	SpiroTiger^®^	Voluntary isocapnic hyperpnea	-	-	6MWT	Improve quality of life and distance in 6MWT
Gonzalez-Montesinos et al.	2020	FeelBreathe^®^	Nasal restriction	18 COPD	8 weeks	PI max y VO_2_max	Positive effects in dynamic hyperinflation, breathing pattern, and breathing efficiency, with higher expiratory and inspiratory time.
Arnedillo et al.	2020	FeelBreathe^®^	Nasal restriction	16 COPD	8 weeks	Inspiratory muscle strength (PImax), dyspnea (mMRC), quality of life (CAT) and exercise capacity (6MWT)	Improvements in quality of life, dyspnea, exercise capacity, and inspiratory muscle strength
Gonzalez-Montesinos et al.	2021	FeelBreathe^®^	Nasal restriction	20 COPD	8 weeks	VO_2_, VCO_2_, respiratory rate	FB added to a pulmonary rehabilitation program in COPD patients could improve tolerance in the incremental exercise test and energy efficiency

COPD: Chronic Obstructive Pulmonary Disease. SGRQ: St. George Respiratory Questionnaire score. BBS: Berg Balance Scale. ABC: activity specific Balance Confidence scale. 6MWT: 6 min walking test. PImax: maximal inspiratory mouth pressure. T,lim: endurance capacity of inspiratory muscles. BQ: Bronchiectasis. FEV_1_: forced expiratory volume in the first second. FVC: forced vital capacity. PImax: maximal inspiratory pressure, PEmax: maximal expiratory pressure. ID: inspiratory duration. SMIP: sustained maximal inspiratory pressure. mMRC: modified Medical Research Council. ISWT: incremental shuttle walk test. ESWT: endurance shuttle walk test. PSQI: Pittsburgh Sleep Quality Index. RR: respiratory rate, VO_2_: oxygen consumption, VCO_2_: carbon dioxide production. FB: FeelBreathe^®^.

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
