# Peer review of "Inspiratory Muscle Training in Patients with Chronic Obstructive Pulmonary Disease (COPD) as Part of a Respiratory Rehabilitation Program Implementation of Mechanical Devices: A Systematic Review"

_ijerph, 2022, doi:10.3390/ijerph19095564_

Round 1
Reviewer 1 Report
The authors present a review regarding COPD and discussing different techniques whether they may influence the outcome of the disease. The manuscript is precisely structured. I suggest to shorten a bit the passage 4 and to write a more detailed figure legend. The conclusion should be more focused on the content of the manuscript. The aims of the review are not presented in detail. Maybe this should be stated in the abstract and introduction sections. A graphical abstract would be nice!
Reviewer 2 Report
A very well presented work is presented. It makes clear what exists in the market in relation to the disease presented.
As suggestions to improve the understanding of the text a little, I suggest the following modifications.
1.-The keywords: to give more power when searching for the article, I would use different words than those used in the title. It would gain power in the usual search engines.
2.-The use of acronyms in some cases would not be justified. Readers unfamiliar with this terminology should be looking up its meaning.
Generate a table of abbreviations as an annex
3.- Table 1. It would be recommendable if it had a sense in the order of appearance of the articles. One possibility would be by seniority or by alphabetical order.
4.- The acronym COPD is used in the table. Although it is the central core of the title, in the caption it is one more acronym and must also appear.
Reviewer 3 Report
The current study may contribute to the field of respiratory rehabilitation in COPD. However, major revisions must be addressed before consideration.
Introduction
1) The introduction is not attractive.
2) The IMT protocol should be better introduced. A brief description for a non-expert audience should be provided.
3)The study's aim is unclear. Is the objective comparing different types of IMT devices? If it is true, why are they supposed different?
Bibliographic review
1) the keywords input in this review is very weak. Relevant studies maybe are not be included because of the poor search strategy.
2) Pubmed is not a database. Why did the authors consider Medline and Pubmed as different databases?
3) A systematic review must be conducted before the next round of peer review.
Round 2
Reviewer 3 Report
Despite the authors' effort to address my comments, a major concern regarding the methodological approach remais unclear.
The inclusion of a systematic method is a notable improvement, but the PRISMA guideline was not correctly followed. In the methodological and results sections: the research strategy should be more detailed, databases choices should be justified (two databases seem to be not enough), inclusion and exclusion criteria should be, the analysis of bias should be provided.
Please see the PRISMA recomendations (http://www.prisma-statement.org), and provide the PRISMA checklist as a supplementary file.
Author Response
The inclusion of a systematic method is a notable improvement, but the PRISMA guideline was not correctly followed.
Firstly, we appreciate the time and effort that you have dedicated to providing your valuable feedback on our manuscript. Regarding the PRISMA recommendations we follow the PRISMA check list 2020 (attached).
In the methodological and results sections: the research strategy should be more detailed, databases choices should be justified (two databases seem to be not enough), inclusion and exclusion criteria should be, the analysis of bias should be provided.
We agree with these comments. Therefore, we have incorporated your suggestions and have rewritten the Methods section to further explain the search strategy, added inclusion and exclusion criteria and included the risk of bias, which was already stated at the Conclusion section.
We look forward to hearing from you in due time regarding our submission and to respond to any further questions and comments you may have.
